# Influence of the Test Method on the Characterization of the Fatigue Delamination Behavior of a Composite Material under Mixed Mode I/II Fracture

**DOI:** 10.3390/polym11111788

**Published:** 2019-11-01

**Authors:** Antonio Argüelles, Clara Rocandio, Silvia Rubiera, Isabel Viña, Jaime Viña

**Affiliations:** 1Department of Construction and Manufacturing Engineering, University of Oviedo, 33203 Gijón, Spain; antonio@uniovi.es (A.A.); rubierasilvia@uniovi.es (S.R.); ivina@uniovi.es (I.V.); 2Department of Materials Science and Metallurgical Engineering, University of Oviedo, 33203 Gijón, Spain; rocandioclara@uniovi.es

**Keywords:** composite, fracture, fatigue, mixed mode I/II

## Abstract

Composite materials manufactured by overlapping plies with certain specific geometries are likely to lose part of their strength due to the presence of internally delaminated regions. The aim of this paper is to experimentally evaluate the generation and propagation of these interlaminar cracks in a carbon-epoxy composite material subjected to fatigue loading under mixed mode I/II fracture. Two different test methods were used for this purpose: The standardized mixed-mode bending (MMB) test and the asymmetric double cantilever beam (ADCB) test, with the goal of exploring the viability of the ADCB test as a simpler alternative to perform than the MMB test, especially in fatigue testing. With this aim in mind and after prior static characterization of the material in which the critical values of the energy release rate were determined under both test methods, the levels of the energy release rate to be applied in fatigue tests were defined for two mode mixity ratios, *G*_II_/*G*_c_ = 0.2 and 0.4 (0.34 ADCB), and a fatigue loading ratio, *R* = *G*_min_/*G*_max_ = 0.1. The G-N fatigue onset curves were subsequently obtained from these experimental data. The most relevant result of the study is that the fatigue limits obtained using the MMB method are generally more conservative than those obtained via the ADCB method.

## 1. Introduction

High performance composite materials are now commonly being used in a wide variety of industries, such as the naval, aeronautical, and sports sectors, among others. The reason for this resides in their excellent mechanical properties, especially their high strength and specific modulus (stiffness). Many polymer matrix composite materials reinforced with continuous fibers are manufactured by successive stacked plies of the material. This configuration means that one of the major faults they may suffer, both during the manufacturing process and throughout their in-service life, is delamination, i.e., the generation of cracks or fissures between two adjacent plies. Said delamination can be propagated under different types of mechanical stresses, both static and dynamic, and can lead to the detachment of adjacent layers of material. The propagation of these interlaminar cracks results in a decrease in the part’s mechanical properties and in some cases may lead to catastrophic failure of the structure. Hence, there is a need to further our knowledge of the damage mechanisms involved in the process and the quantification of the strength of the material subject to this phenomenon. Numerous test methods have been developed for this purpose, all of which are based on fracture mechanics, in pure mode fracture [1,2,3,4] as well as mixed mode fracture [5,6,7,8,9]. The most widely used mixed mode I/II test in the field standardized by ASTM is known as the mixed-mode bending (MMB) test [10]. However, other methods have been developed in which the test device is less complex, among which is the asymmetric double cantilever beam (ADCB) test. This test has the advantage of presenting a simple test configuration, similar to that used in the characterization of the material under mode I fracture, using very simple test devices. The difference between the two is that in the ADCB specimen, the crack lies outside the plane of symmetry. This asymmetry originates mixed mode I/II fracture in the crack front and different mode mixity ratios can be obtained by varying the position of the crack with respect to its midplane. Some of the first studies performed in this field using this test setup were carried out by Bao et al. [11], in which the authors put forward a formulation that allows the total energy release rate and its partitions, components of modes I and II fracture, to be obtained taking into account the orthotropic characteristics of the material. Qiao et al. [12] modelled several types of beam geometries depending on the types of joint in the crack front. Subsequently, Ducept et al. [13] developed two calculation methods and the partition of the energy release rate into its corresponding modes I and II fracture: The global and local method. While the global method was only able to predict the behavior in mode I fracture, it was observed that the local method provided good results in the determination of G_I_ and G_II_ and was subsequently used and adapted by various authors, such as Prombut et al. [14]. Mollón et al. [15,16] developed an adaptation of the modified beam theory for this test by means of a flexibility calibration, from which the adjustment parameters were obtained. In turn, Shokrieh et al. [17] developed the modified elastic beam ttheory (MEBT) method from that proposed by Mollón et al.

Currently, this test method continues to be used with the aim of further developing its potential applications [18,19,20,21,22,23,24]. However, the fact that most of the failures of materials destined for industrial use are due to fatigue phenomena means that it is necessary to carry out a study of the delamination process under dynamic loading [25,26,27,28], as some authors have proposed for different types of tests.

The present paper analyses the fatigue behavior of an epoxy carbon composite material subjected to mixed mode I/II fracture, using two test methods (ADCB and MMB) for two mode mixity ratios in order to assess the potential of the ADCB test as a substitute for the MMB test method in this type of fatigue testing.

## 2. Materials and Test Specimens

The material studied in this work was manufactured from an epoxy resin prepreg with an MTM45-1/IM7 (12k)-134 gr/m^2^ unidirectional carbon reinforcement and 32% resin by weight. An oven with the thermal cycle recommended by the maker was used for vacuum molding. Table 1 shows the elastic moduli (E and G) in the longitudinal direction of the fibers (“11”) and in the transverse direction (“22”), as well as the normal and shear ultimate strengths (σ, τ).

This composite was manufactured via successive manual stacking of prepreg layers. A non-adhesive insert generating the initial delamination was introduced during this process (see Figure 1). As a function of the test method, it was located in the mid-plane of the laminate for MMB tests and outside the mid-plane for ADCB tests, varying the mode mixity ratio by varying the asymmetrical positioning of the insert: For G_II_/G_T_ = 0.2: [0°_15_//0°_45_] and for G_II_/G_T_ = 0.34: [0°_7_//0°_53_]. It should be noted that although the initial aim of the study was to compare two previously defined mode mixity ratios (0.2 and 0.4), the mode mixity ratio in the case of the ADCB tests is limited by the specimen geometry and it was not possible to achieve a ratio of 0.4 for this material. Hence, the maximum achievable ratio, 0.34, was finally used in this study. This does not substantially affect the conclusions obtained from the study.

## 3. Experimental Method

All the fatigue tests, both static and dynamic, were performed on an MTS Model 810 servohydraulic test machine equipped with a 5-kN load cell. The two test methods used in the study are schematically represented in Figure 2, which allows the differences between the two test methods to be clearly seen.

### 3.1. Characterization under Static Loading

The critical fracture properties of the material were determined under static loading: Loads, displacements, and critical energy release rates (*P*_c_, *δ*_c_, *G*_c_). In order to determine the energy release rate, the formulations proposed by ASTM D6671/D6671M-13E1 [10] were used for the MMB test typology while the formulation proposed by Mollón et al. [15,16] was employed in the case of the ADCB test, in which the total energy release rate (*G*_c_ = *G*_I_ + *G*_II_) is obtained by means of the following expression:(1)$$G_{c}=\frac{3P^{2}{\left(a+\Delta \right)}^{2}}{2BEI_{eq}}$$

The parameters Δ and *EI_eq_* are obtained via calibration of the flexibility from the graphical representation of the cube root of the flexibility versus crack length, where Δ represents the intersection with the abscissa axis and (1/EI_eq_)^1/3^ is the slope of the calibration line. The mode mixity ratio can be calculated via the expression:(2)$$\frac{G_{II}}{G_{C}}=-T\sqrt{1-\alpha^{2}}+T$$
in which α is a function of the thickness ratio (*h*_1_ and *h*_2_), according to the expression:(3)$$\alpha =\frac{1-\frac{h_{1}^{3}}{h_{2}^{3}}}{1+\frac{h_{1}^{3}}{h_{2}^{3}}}$$

*T* is an adjustment parameter:(4)T(α)=0.06α+0.35

### 3.2. Fatigue Characterization

The objective of the experimental fatigue program was to determine the fatigue curves of the tested material when subjected to fracture delamination processes under dynamic mixed mode I/II loading. In this study, fatigue failure of the material was considered to occur at the onset of propagation of an interlaminar crack in the test specimen.

These tests were performed at constant levels of loading for each test method in combination with isolated tests. To define these levels, the results obtained from the prior characterization of the material under static loading were taken as a reference, calculating them as percentages of the critical energy release rate, *G*_c_. All fatigue tests were performed with an asymmetry coefficient of *R* = *G*_min_/*G*_max_ = 0.1.

## 4. Experimental Results and Discussion

The experimental results are presented below.

### 4.1. Static Loading

Table 2 and Table 3 present the results for the two mode mixity ratios studied here under static loading using the MMB and ADCB test methods. The total energy release rate, *G*_c_, and its partitions are presented: Component in mode I, *G*_I_, and component in mode II, *G*_II_, as well as the mode mixity ratio obtained, *G*_II_/*G*_c_.

It can be seen that as the proportion in mode II fracture increases, the total energy release rate increases, a finding consistent with the behavior of the material in pure modes I and II fracture.

For the ADCB test method, the energy release also increases as the mode mixity ratio increases.

For the two mode mixity ratios considered here, it is observed that the values of the total energy release rate obtained using the ADCB method are higher than those obtained with the MMB method. For a mode mixity ratio of 0.2, higher values of the energy release rate of around 37% are obtained for the ADCB method while for a ratio of 0.4 (performing a linear extrapolation of values for ADCB testing), the values are around 21%. That is to say, this trend is somewhat more pronounced in the lower mode mixity ratio.

### 4.2. Dynamic Loading, Fatigue Life

In order to improve the reliability of the evaluation of the results obtained from the experimental program, it was considered convenient to conduct a probabilistic analysis of the entire fatigue lifespan, for which purpose different models exist [29,30,31]. In this study, we used a model based on a Weibull distribution proposed by Castillo et al. [32,33] that allows normalization of the entire fatigue lifespan and which has already been shown to be effective in other cases involving composite materials [34].

Figure 3 and Figure 4 show the fatigue behavior of the material under mixed mode I/II loading using the MMB test method for the two mode mixity ratios considered in this paper (0.2 and 0.4) and fatigue failure probabilities of 5% and 50%, respectively. The total energy release rate: *G*_total_ = (*G*_I_+*G*_II_) applied to the test specimens was plotted against the number of cycles supported during the fatigue test.

For the MMB test and the two probabilities of fatigue failure, the same trend is observed for the higher mode mixity ratio (0.4). In the low- and medium-number-of-cycles regions, relatively low fracture energies are produced compared to the values obtained in the prior static characterization of the material (around 27% *G*_c_). This seems to indicate that the dominant parameter in the fatigue failure process is the proportion of mode II fracture applied to the tested specimens. This trend persists in the infinite lifespan region, though to a somewhat less pronounced degree.

Figure 5 and Figure 6 show the fatigue behavior of the material under mixed mode I/II loading using the ADCB test method for the two mode mixity ratios considered (0.2 and 0.34) and fatigue failure probabilities of 5% and 50%, respectively.

For the material considered in this study subjected to fatigue loading using the ADCB test method, the observed trends are as follows. In the low-number-of-cycles region, fatigue failure occurs at load levels of 68% *G*_c_ for a mode mixity ratio of 0.2 while for a ratio of 0.4, the levels are around 50% *G*_c_, though for a higher number of fatigue life cycles. In the infinite lifespan region, fatigue limits of 16% and 21% *G*_c_ are observed for a mode mixity ratio of 0.2 and failure probability of 5% and 50%, respectively. While for a mode mixity ratio of 0.34, the fatigue limits are 12.5% for a 5% failure probability and 17% for 50% failure probability. Thus, from the point of view of their fatigue life, in the infinite lifespan region, there is a greater loss in material strength as the proportion of mode II failure increases.

Figure 7 and Figure 8 for both test methods and mode mixity ratios *G*_II_/*G*_c_ = 0.2 and 0.4 (0.34 for the ADCB test) respectively show the plots representing the experimental data together with the fatigue failure probability, 5% and 50% according to the Weibull model referenced above, in which the total fracture energy is represented. This value represents the maximum energy release rate (*G*_I_ + *G*_II_) applied to the test specimens during fatigue tests under mixed mode I/II loading versus the number of cycles required for the onset of a fatigue crack. It should be recalled that, in the present study, the specimen was considered to have failed due to fatigue when the onset of the delamination process, identified by a change in flexibility of the specimen, occurs and is visually appreciated on one of the sides of the test specimen.

As to the fatigue behavior of the material, depending on the test method employed, MMB or ADCB, different behavior is observed in the low-number-of-cycles and high-number-of-cycles regions when the mode mixity ratios applied to the test specimens are modified. For the failure probabilities considered in this study, the maximum fracture energy for a mode mixity ratio of 0.4 obtained in the MMB test is significantly lower than that obtained for a mode mixity ratio of 0.2. Whereas for the ADCB test, the fracture energies obtained in the low-number-of-cycles region are similar for the two mode mixity ratios considered.

As for the infinite lifespan region, the fatigue limit considered in this study corresponded to an experimental level exceeding two million cycles. The behavior of the material can be known up to 10 million cycles using the aforementioned probabilistic model. Different behaviors are observed depending on the mode mixity ratio employed. For a mode mixity ratio of 0.4, similar fatigue limits of around 60 J/m^2^ are obtained for both types of test method and both failure probabilities. For a mode mixity ratio of 0.2, the fatigue limit is conditioned by the test method used in the fatigue characterization of the material, resulting in fatigue limits of 36 J/m^2^ for the MMB method, and 61 J/m^2^ for the ADCB method.

## 5. Analysis of Fracture Surfaces

Although there was no previous control of the existence of internal defects or delaminations, after the test, the surfaces of all the specimens were visually analyzed and no defects that could affect delamination were observed in the main plane. The most relevant aspects observed in the analysis of the fracture surfaces of the specimens tested to fatigue under mixed mode I/II loading using the MMB and ADCB test methods are presented below, with the aim of identifying the dominant fracture processes. A JEOL 6610LV scanning electron microscope (SEM) was used for this purpose [35,36]. Starting with the specimens tested by means of the MMB method, Figure 9 shows the region close to the insert used to initiate delamination in a fatigue specimen for a mode mixity ratio of 0.2. The existence of broken fibers can be observed as proof of the prior existence of fiber bridges [37,38,39] and also river markings [37,40], characteristic structures of mode I fracture, together with cusps [39,41,42,43], which are characteristic of mode II fracture. This clearly indicates that both fracture modes coexist at the crack front when carrying out the test.

When analyzing the counter-specimen, a phenomenon that appears on the opposite side of the river markings was detected, which can be seen in Figure 10. This figure shows how the river markings leave what could be called as tree roots in the counterspecimen.

Figure 11 shows the fracture surface of a specimen tested by means of the ADCB method with a mode mixity ratio of 0.2 under fatigue loading. The observed microstructures are very similar, though with small variations. Broken fibers as well as river markings and cusps can be seen in the same image, demonstrating the simultaneous presence of characteristic morphologies of modes I and II fracture. The river markings and cusps can also be seen under greater magnification in Figure 12.

An important morphology that differentiates both test methods and one that has been detected in ADCB specimens tested to failure, but not in MMB specimens, is the presence of stretch marks, which clearly show the onset of fatigue-generated crack growth. It can be seen that the size of the observed vertical markings is different, which is an unmistakable sign of the instability of the initial growth of fatigue delamination. Specifically, in this case, these findings can be attributed to the crack growing faster on the left side of the image than on the right (Figure 13).

## 6. Conclusions

The most significant conclusions regarding the influence of the test method (MMB or ADCB) on the fatigue behavior of a composite material submitted to mixed I/II loading for two mode mixity ratios are presented below.

Regarding the experimental results for the two test methods under static loading, slightly higher values were obtained for the two mode mixity ratios studied in this paper when the trials were performed using the ADCB test. Fatigue characterization of a composite material under mixed mode I/II fatigue loading is simpler from the experimental point of view using the ADCB test method, although the application of this method is conditioned by the desired mode mixity ratio under which the characterization of the composite material is to be carried out.

As regards fatigue behavior, for the higher mode mixity ratio considered in this study, 0.4 for the MMB test and 0.34 for the ADCB test, the obtained fatigue limits were practically the same. Thus, the ADCB test could be used as a substitute for the MMB test to determine the fatigue limit of the material used in this study. However, for a low mode mixity ratio, 0.2 in this study, the above conclusion could not be decisively drawn in view of the results provided by the MMB method, which are significantly lower than those provided by the ADCB method.

For the ADCB test, the obtained fatigue limit does not depend on the mode mixity ratio employed, although the dispersion of fatigue results is greater when the proportion of mode II fracture increases. In general, the dispersion of the experimental results is greater for the ADCB test method than for the MMB method and for both of the mode mixity ratios studied in this paper.

The following conclusions can be drawn from the fractographic failure analysis: There is no representative difference between the fracture surfaces under mixed mode static or dynamic loading, especially when using the MMB test method. The existence of differences between the surfaces obtained in tests carried out with different percentages of modes I and II fracture is negligible. Under mixed mode loading using the ADCB test method, specific morphologies can be observed that represent a certain crack growth during the onset of delamination before a representative change in the flexibility of the specimen occurs, allowing its detection during the test phase. Apart from this particular morphology, the fracture surfaces obtained for the MMB and ADCB test methods are similar.

## Figures and Tables

**Figure 1 polymers-11-01788-f001:**
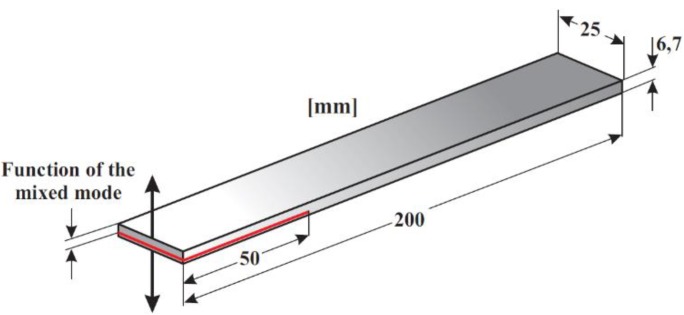
Geometry of the specimen.

**Figure 2 polymers-11-01788-f002:**
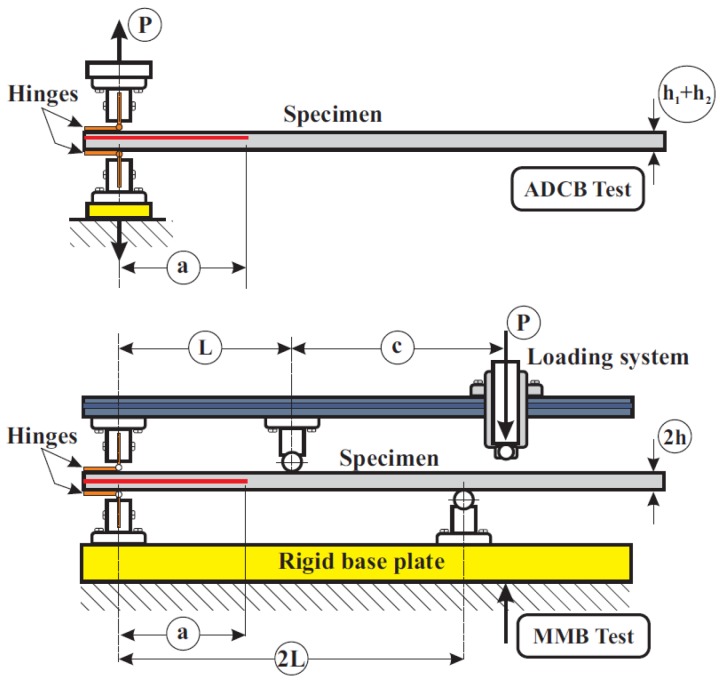
Scheme of the ADCB and MMB test methods employed.

**Figure 3 polymers-11-01788-f003:**
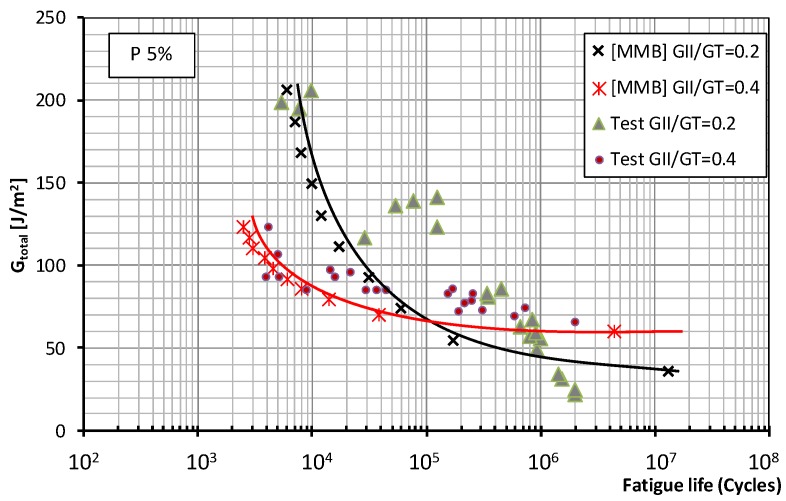
Material fatigue curves, MMB test, for a 5% probability of fracture and two mode mixity ratios: 0.2 and 0.4.

**Figure 4 polymers-11-01788-f004:**
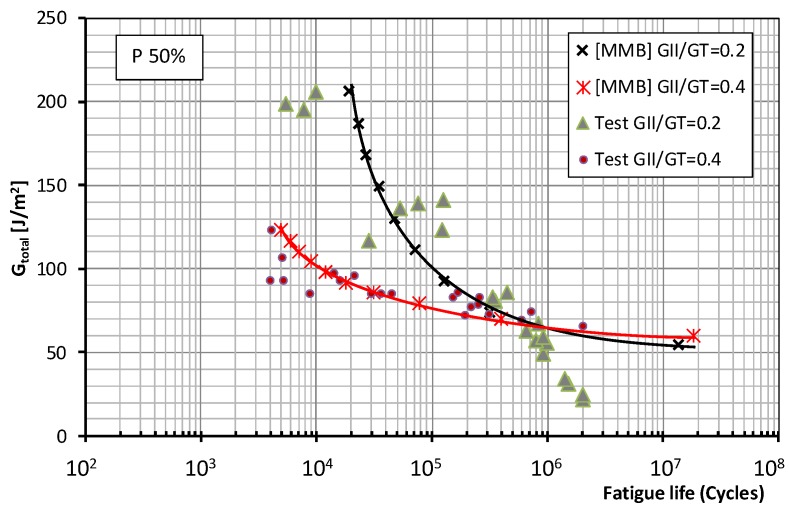
Material fatigue curves, MMB test, for a 50% probability of fracture and two mode mixity ratios: 0.2 and 0.4.

**Figure 5 polymers-11-01788-f005:**
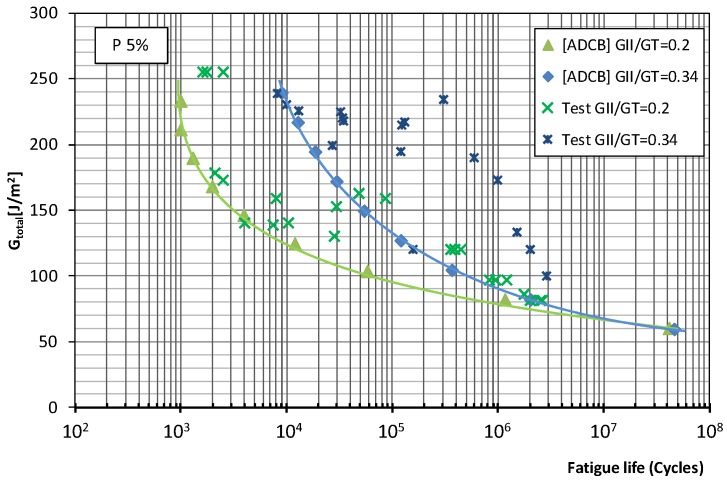
Material fatigue curves, ADCB test, for a 5% probability of fracture and two mode mixity ratios: 0.2 and 0.34.

**Figure 6 polymers-11-01788-f006:**
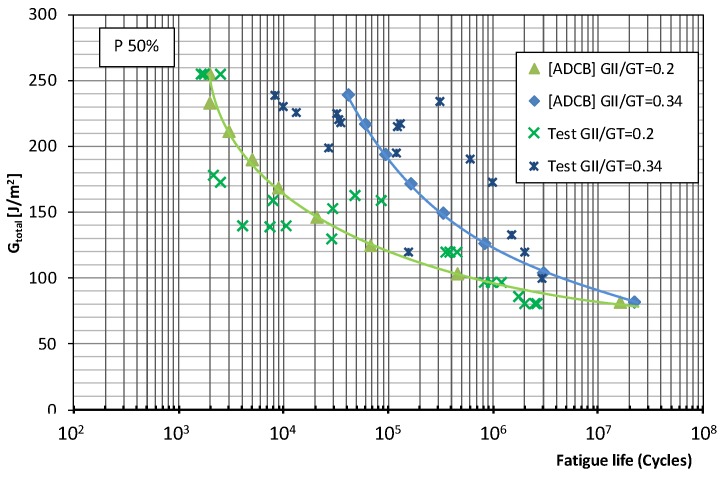
Material fatigue curves, ADCB test, for a 50% probability of fracture and two mode mixity ratios: 0.2 and 0.34.

**Figure 7 polymers-11-01788-f007:**
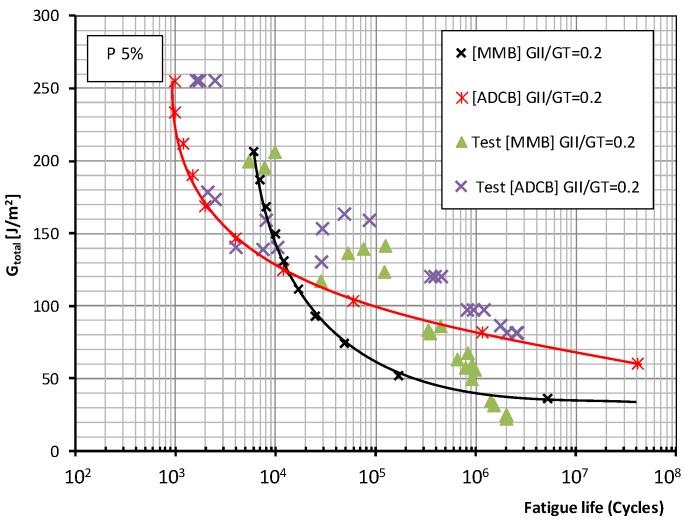
Fatigue behavior for a mode mixity ratio (*G*_II_/*G*_C_) of 0.2, a 5% failure probability, and the MMB and ADCB test methods.

**Figure 8 polymers-11-01788-f008:**
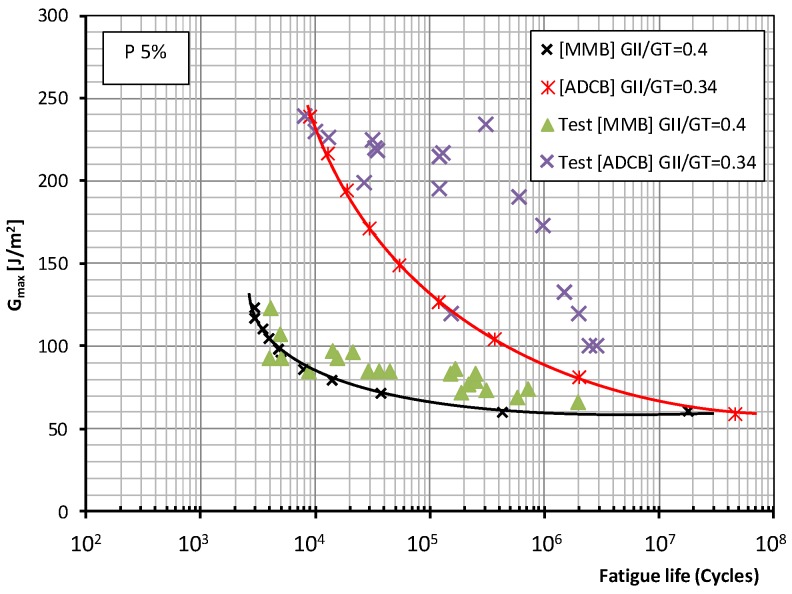
Fatigue behavior for a mode mixity ratio (*G*_II_/*G*_C_) of 0.4 for the MMB test and of 0.34 for the ADCB test and a 5% failure probability.

**Figure 9 polymers-11-01788-f009:**
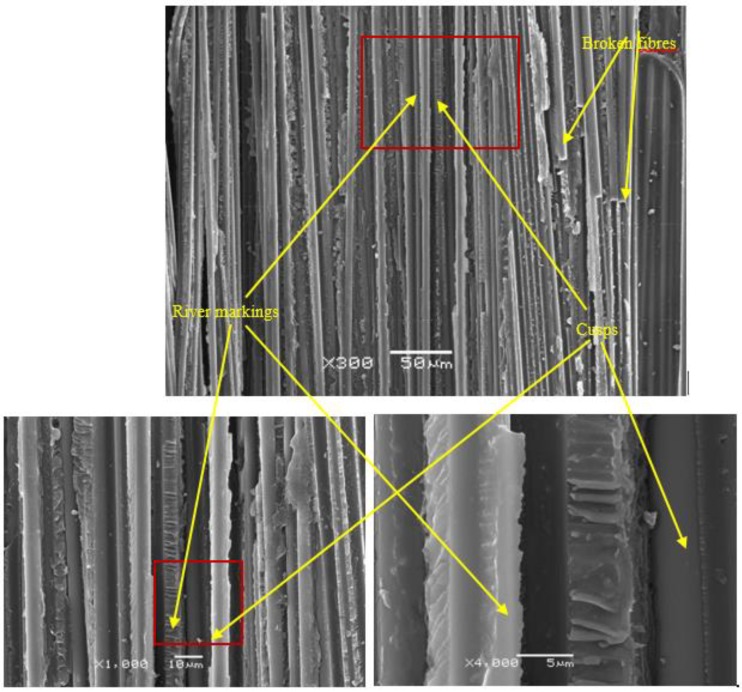
SEM micrographs of a fatigue specimen tested with the MMB device showing the presence of cusps, river markings, and broken fibers.

**Figure 10 polymers-11-01788-f010:**
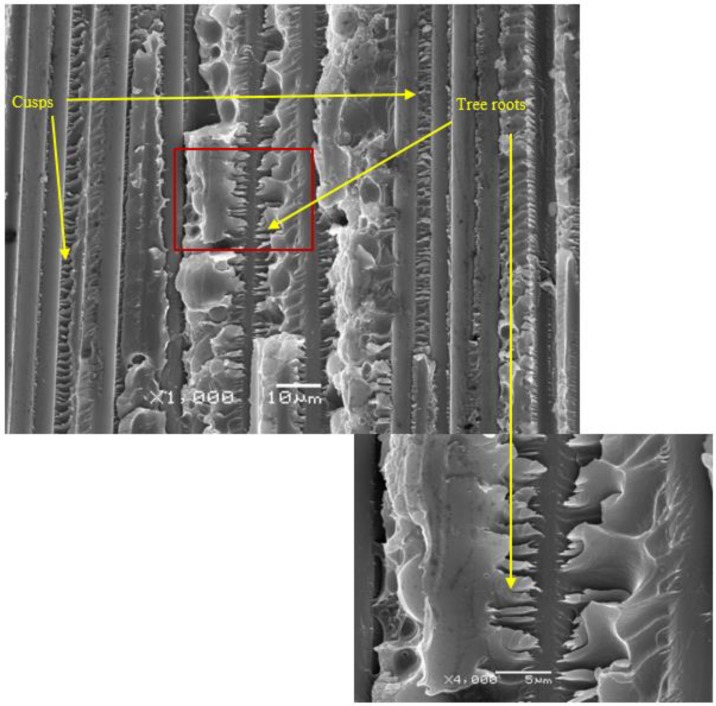
SEM micrographs of a fatigue specimen (counterspecimen in Figure 9) tested with the MMB device showing the presence of cusps and tree roots.

**Figure 11 polymers-11-01788-f011:**
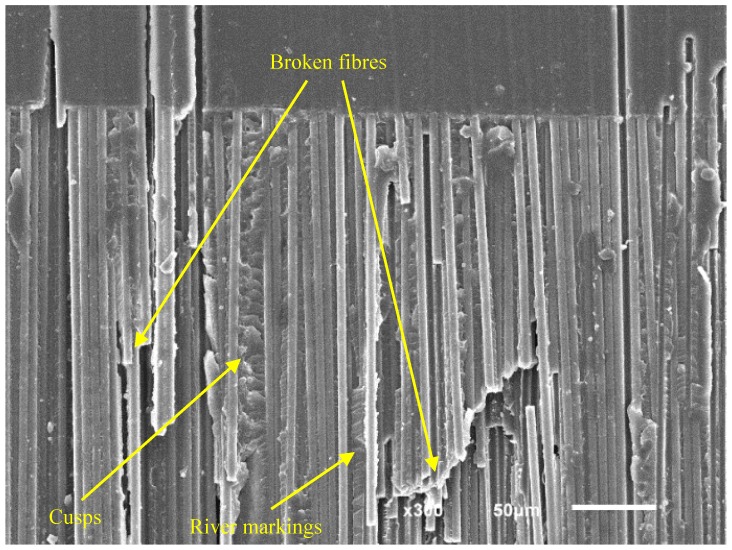
SEM micrographs of an ADCB fatigue specimen showing the presence of broken fibers, cusps, and river markings.

**Figure 12 polymers-11-01788-f012:**
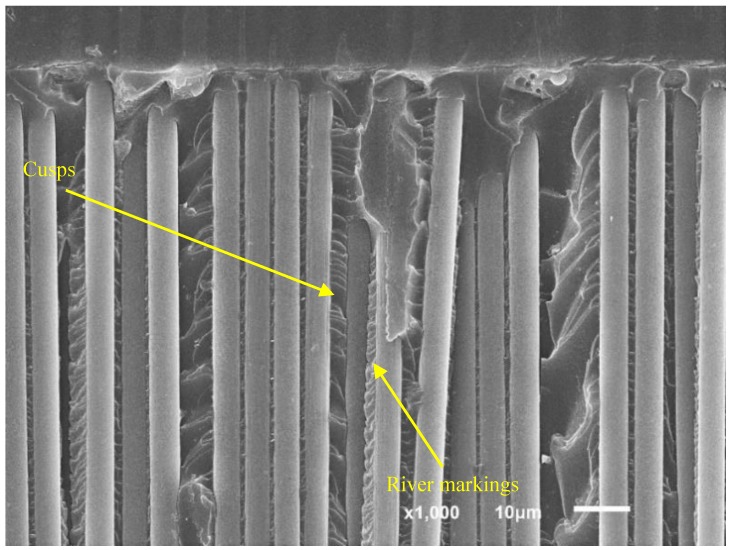
SEM micrographs of an ADCB fatigue specimen showing the presence of cusps and river markings.

**Figure 13 polymers-11-01788-f013:**
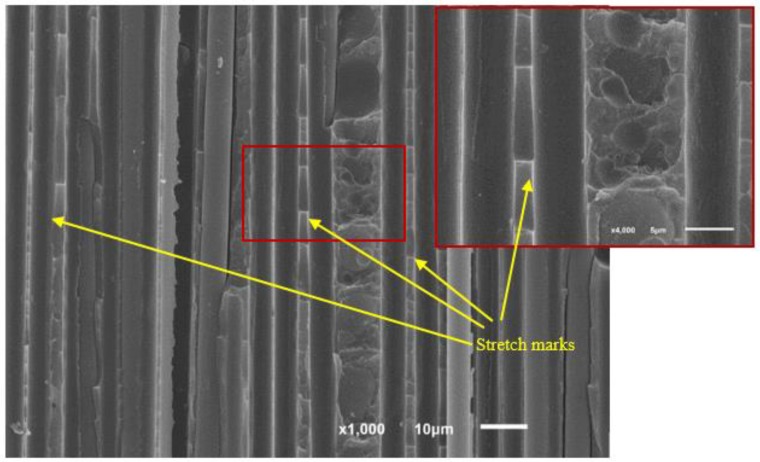
SEM micrographs of an ADCB fatigue specimen showing the presence of cusps and stretch marks.

**Table 1 polymers-11-01788-t001:** Mechanical properties of the material.

Elastic Moduli		Ultimate Tensile Stresses		Shear Modulus	Shear Strength
E_11_ (GPa)	E_22_ (GPa)	σ_11_ (MPa)	σ_22_ (MPa)	G_12_ (GPa)	τ_max_ (MPa)
174	13	2199	36	5.3	93

**Table 2 polymers-11-01788-t002:** MMB results, static loading.

	*G*_c_ (J/m^2^)		*G*_I_ (J/m^2^)		*G*_II_ (J/m^2^)	
*G*_II_/*G*_c_	0.2	0.4	0.2	0.4	0.2	0.4
Mean	271	469	216	280	54	189
S.D.	23	36	19	21	4	15
C.V. [%]	8.5	7.6	8.8	7.5	7.4	7.9

**Table 3 polymers-11-01788-t003:** ACDB results, static loading.

	*G*_c_ (J/m^2^)		*G*_I_ (J/m^2^)		*G*_II_ (J/m^2^)	
*G*_II_/*G*_c_	0.2	0.34	0.2	0.34	0.2	0.34
Mean	372	480	285	320	87	161
S.D.	25	108	32	21	7	15
C.V. [%]	6.7	22.5	11.2	6.5	8.0	9.3

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
