# Peer review of "Influence of the Test Method on the Characterization of the Fatigue Delamination Behavior of a Composite Material under Mixed Mode I/II Fracture"

_polymers, 2019, doi:10.3390/polym11111788_

Round 1

Reviewer 1 Report

The manuscript entitled "Influence of the Test Method on the Characterization of the Fatigue Delamination Behaviour of a Composite Material under Mixed Mode I/II Fracture" deals with the characterization of the generation and propagation of cracks in a carbon/epoxy composite through two different test methods. 

In my opinion, the topic of the manuscript could be interesting for the potential readers of Polymers, therefore I suggest the publication of the paper, given that the reported data are well organized and the conclusions are supported by the obtained results.

I suggest to the authors to verify the English language of the abstract part, since at line 18 the meaning of the sentence is not clear. Furthermore, I suggest to improve the quality of the expressions present in section 3.1 

Author Response

Reviewer 1

The manuscript entitled "Influence of the Test Method on the Characterization of the Fatigue Delamination Behaviour of a Composite Material under Mixed Mode I/II Fracture" deals with the characterization of the generation and propagation of cracks in a carbon/epoxy composite through two different test methods. 

In my opinion, the topic of the manuscript could be interesting for the potential readers of Polymers, therefore I suggest the publication of the paper, given that the reported data are well organized and the conclusions are supported by the obtained results.

I suggest to the authors to verify the English language of the abstract part, since at line 18 the meaning of the sentence is not clear.

A modification was made in order to clarify the sentence

Furthermore, I suggest to improve the quality of the expressions present in section 3.1 

The quality of the expressions was improved

Reviewer 2 Report

This manuscript represents the experimental evaluation of interlaminar cracks due to fatigue loading subjected to mixed mode I/II fracture in CFRP laminates. Two different standard tests were carried out: MMB test and ADCB test. The results of both tests were compared. The structure of the paper is well organized and the overall paper tells a logical story with a concrete conclusion. It is suitable for publication in Polymers after minor revision:

1) Please explain in more detail the manufacturing process of the two types of specimens: RTM, hot-pressing, autoclave, ...

2) Please explain if the manufactured quality of the different specimens has been analyzed to avoid internal defects as delaminations using NDT techniques or others (Ultrasounds, ...) 

3) The quality of the different equations must be improved and they must be numbered

Author Response

Reviewer 2

This manuscript represents the experimental evaluation of interlaminar cracks due to fatigue loading subjected to mixed mode I/II fracture in CFRP laminates. Two different standard tests were carried out: MMB test and ADCB test. The results of both tests were compared. The structure of the paper is well organized and the overall paper tells a logical story with a concrete conclusion. It is suitable for publication in Polymers after minor revision:

1) Please explain in more detail the manufacturing process of the two types of specimens: RTM, hot-pressing, autoclave, ...

The manufacturing process has been detailed

2) Please explain if the manufactured quality of the different specimens has been analyzed to avoid internal defects as delaminations using NDT techniques or others (Ultrasounds, ...) 

In the text has been explained the visual control carried out

3) The quality of the different equations must be improved and they must be numbered

The quality of the equations was improved
